

# Modeling the effects of tropospheric ozone on the growth and yield of global staple crops with DSSAT v4.8.0

Jose Rafael Guarin[1,2,*], Jonas Jägermeyr[1,2], Elizabeth A. Ainsworth[3], Fabio A. A. Oliveira[4], Senthold Asseng[5], Kenneth Boote[4], Joshua Elliott[6], Lisa Emberson[7], Ian Foster[8], Gerrit Hoogenboom[4], David Kelly[8], Alex C. Ruane[2], Katrina Sharps[9]

[1]Center for Climate Systems Research, Columbia Climate School, Columbia University, New York, NY 10025 USA
[2]NASA Goddard Institute for Space Studies, New York, NY 10025 USA
[3]Global Change and Photosynthesis Research Unit, United States Department of Agriculture, Agricultural Research Service, Urbana, IL 61801
[4]Department of Agricultural and Biological Engineering, University of Florida, Gainesville, FL 32611 USA
[5]School of Life Sciences, HEF World Agricultural Systems Center, Technical University of Munich, Freising, 85354 Germany
[6]Center for Robust Decision-making on Climate and Energy Policy (RDCEP), University of Chicago, Chicago, IL 60637 USA
[7]Environment & Geography Dept., University of York, York, YO10 5NG UK
[8]Department of Computer Science, University of Chicago, Chicago, IL 60637 USA
[9]UK Centre for Ecology & Hydrology, Environment Centre Wales, Bangor, LL57 2UW, UK
*Correspondence to: Jose Rafael Guarin (j.guarin@columbia.edu)

**Highlights**

• Effects of $O_3$ stress on photosynthesis and leaf senescence were added to the DSSAT/pDSSAT maize, rice, soybean, and wheat crop models.

•  The modified models reproduced growth and yields under different $O_3$ levels observed in field experiments and reported in the literature.

•  Expected detrimental interactions between $O_3$, $CO_2$, and water deficit were reproduced with the new models.

• The updated crop models can be used to simulate impacts of $O_3$ stress under future climate change and air pollution scenarios.

**Abstract.** Elevated surface ozone ($O_3$) concentrations can negatively impact growth and development of crop production by reducing photosynthesis and accelerating leaf senescence. Under unabated climate change, future global $O_3$ concentrations are expected to increase in many regions, adding additional challenges to global agricultural

production. Presently, few global process-based crop models consider the effects of $O_3$ stress on crop growth. Here, we incorporated the effects of $O_3$ stress on photosynthesis and leaf senescence into the Decision Support System for Agrotechnology Transfer (DSSAT) crop models for maize, rice, soybean, and wheat. The advanced models reproduced the reported yield declines from observed $O_3$-dose field experiments and $O_3$ exposure responses reported in the literature ($O_3$ relative yield loss RMSE < 10% across all calibrated models). Simulated crop yields decreased as

daily $O_3$ concentrations increased above 25 ppb, with average yield losses of 0.16% to 0.82% (maize), 0.05% to 0.63% (rice), 0.36% to 0.96% (soybean), and 0.26% to 1.23% (wheat) per ppb $O_3$ increase, depending on the cultivar $O_3$ sensitivity. Increased water deficit stress and elevated $CO_2$ lessen the negative impact of elevated $O_3$ on crop yield,



but potential yield gains from $CO_2$ concentration increases may be counteracted by higher $O_3$ concentrations in the future, a potentially important constraint to global change projections for the latest process-based crop models. The improved DSSAT models with $O_3$ representation simulate the effects of $O_3$ stress on crop growth and yield in interaction with other growth factors and can be run in the parallel DSSAT global gridded modeling framework for future studies on $O_3$ impacts under climate change and air pollution scenarios across agroecosystems globally.

**Keywords**

Surface ozone, climate change, global process-based crop model, phytotoxicity, staple crop yield

## 1 Introduction

Surface or ground-level, ozone ($O_3$) is a major air pollutant that causes adverse impacts on agricultural productivity worldwide (Mills et al., 2018b; Emberson et al., 2018; Tai et al., 2021). $O_3$ is formed through photochemical reactions between incoming solar radiation and primary pollutants such as Nitrogen Oxides ($NO_x = NO + NO_2$), Volatile Organic Compounds (VOCs), Carbon Monoxide (CO), or Methane ($CH_4$) across all areas of the globe (Cooper et al., 2014; Simpson et al., 2014). Global $O_3$ concentrations have increased 2-7% per decade in northern mid-latitude regions and 2-12% per decade in tropical regions since the mid-1990s (Ipcc, 2021; Arias et al., 2021). Future $O_3$ concentrations are projected to continue increasing if $O_3$ precursor emissions are not mitigated, i.e., following the shared socio-economic pathways where regional rivalry leads to doubling of $CO_2$ emissions by 2100 (SSP3-7.0) or where fossil fuel enabled growth leads to doubling of $CO_2$ emissions by 2050 (SSP5-8.5) (Ipcc, 2021; Arias et al., 2021; Szopa et al., 2021; Griffiths et al., 2021).

Crops exposed to elevated levels of $O_3$ concentrations can experience reduced photosynthesis, accelerated senescence, foliar chlorosis and even necrosis from increased cumulative oxidative stress (Ainsworth, 2017). These negative effects lead to decreased productivity resulting in global yield losses between 2-16% for the four main staple crops: maize, rice, soybean, and wheat (Ainsworth, 2017; Schiferl and Heald, 2018; Emberson, 2020), with global annual economic damages of approximately \$34 billion (Sampedro et al., 2020; Feng et al., 2022). Climate change may exacerbate the negative effects from elevated $O_3$ concentrations because $O_3$ concentrations are highest in summer months and the projected higher temperatures with more frequent heat waves may lead to a longer period of more active photochemical reactions (Zhang and Wang, 2016; Hou and Wu, 2016; Szopa et al., 2021). Elevated concentrations of atmospheric $CO_2$ and increased periods of water deficit stress cause stomatal closure that can reduce crop $O_3$ uptake (Khan and Soja, 2003; Biswas et al., 2013), but in turn potential yield gains associated with the $CO_2$ fertilization effect (Toreti et al., 2020; Jagermeyr et al., 2021) may be constrained by elevated $O_3$. Therefore, it is important to evaluate net $O_3$ effects for crop growth and consider the effects of $O_3$ in global agricultural assessments examining future scenarios.

Process-based crop simulation models have been used to evaluate the impacts of $O_3$ on crop yields (Guarin et al., 2019; Tai et al., 2021), but most global gridded process-based crop models are still unable to respond to $O_3$ stress. Recently, the global Lund-Potsdam-Jena managed Land (LPJmL) and Joint UK Land Environment Simulator



(JULES) models were modified to include the effects of $O_3$ stress on soybean and wheat growth (Schauberger et al., 2019; Leung et al., 2020). Additionally, the Agricultural Model Intercomparison and Improvement Project (AgMIP; Rosenzweig et al. (2013)) Ozone Team has recently developed protocols for incorporating $O_3$ stress into a wider body of crop models aiming to establish the first multi-model assessment of ozone impacts in agriculture at global level (Emberson et al., 2018).

The aim of this study is to incorporate the effects of $O_3$ concentrations into the stress response functions of the maize, rice, soybean, and wheat models within the established Decision Support System for Agrotechnology Transfer (DSSAT) v4.8.0 modeling platform (Jones et al., 2003; Hoogenboom et al., 2019), and consequently the parallel DSSAT (pDSSAT) v4.8.0 global gridded modeling platform that is used to run DSSAT in a global setup (Elliott et al., 2014), to simulate $O_3$ effects on global crop development and yield for the four major staple crops. The observational data from the Free-air $CO_2$ Enrichment (FACE) field experiments conducted in Champaign, Illinois, USA (Choquette et al., 2020; Betzelberger et al., 2012) and well-known $O_3$ exposure relationships reported in the literature are used to develop and calibrate the model $O_3$ response functions. Additionally, the observed interactions between $O_3$, $CO_2$, and water deficit stress are examined via sensitivity analyses conducted with the modified models.

## 2 Materials and Methods

### 2.1 Description of crop models

The crop models within the pDSSAT parallel modeling environment are based on the existing crop models within the widely used DSSAT crop modeling platform (Jones et al., 2003; Hoogenboom et al., 2019) combined with the Center for Robust-Decision Making on Climate and Energy Policy (RDCEP) Parallel System for Integrating Impact Models and Sectors (pSIMS) framework (Elliott et al., 2014) to allow for global gridded process-based crop modeling on high performance computational systems. The $O_3$ stress routines presented here are also applied in the standard DSSAT crop models and can be used for field level simulations and point-based testing in addition to the global-level modeling applications.

The four DSSAT crop models used in this study are the Crop Environment Resource Synthesis (CERES) -Maize, CERES-Rice, Crop Growth Simulation (CROPGRO) -Soybean, and Nitrogen Wheat (NWheat) models that have been used in previous AgMIP crop model intercomparisons (Bassu et al., 2014; Li et al., 2015; Asseng et al., 2015; Kothari et al., 2022). The CERES-Maize and CERES-Rice models were previously used to estimate global ozone crop losses (Schiferl and Heald, 2018); however, their approach was based on the multiplication of the simulated global base production by the relative yield-$O_3$ response functions to determine a response proxy. The approach used in this present study integrates daily process-based stress calculations to simulate daily crop growth and stress dynamics. Thus, the models are more applicable to a much broader range of scenarios given that they can combine daily stress interactions and can be used to scale across agroecosystems in a more robust way.

### 2.2 $O_3$ incorporation into the crop models





The incorporation of $O_3$ effects into the DSSAT crop models followed the same methodology as the $O_3$ incorporation into the DSSAT-NWheat crop model (Guarin et al., 2019). $O_3$ response was added to the models via the inclusion of daily photosynthesis reduction and leaf senescence acceleration functions. Additionally, the interaction between $O_3$ and water deficit stress and/or atmospheric $CO_2$ concentrations was incorporated into the models since these combined interactions can mitigate impacts from $O_3$ on crop production and vice-versa. For example, water deficit stress that

induces stomatal closure in turn limits $O_3$ stress because of reduced aerosol uptake (Khan and Soja, 2003; Biswas et al., 2013).

### 2.2.1 CERES-Maize and CERES-Rice models

The effects of $O_3$ were incorporated into the CERES-Maize and CERES-Rice models using similar methodology since these two models share similar code. $O_3$ was added into the models using a photosynthesis reduction stress factor

($FO_3$) following Eq. (1):

$$FO_3 = \max\left(0.0, -\left(\frac{FOZ_1}{100}\right) * OZON_7 + \left(1.0 + \left(\frac{FOZ_1}{100}\right) * 25.0\right)\right),$$      (1)

where $OZON_7$ is the daily mean 7-hour (M7, 9:00 – 15:59 hr) $O_3$ concentration (ppb) and $FOZ_1$ is the $O_3$ stress parameter for photosynthesis calibrated for different $O_3$ sensitivities of cultivars divided by a decimal correction factor of 100. The decimal correction factor ensures that the $FOZ_1$ parameter value ranges between 0.0 and 1.0 in the model

ecotype parameter file for comprehensible user input. A minimum M7 $O_3$ threshold of 25 ppb was set as the reference value based on pre-industrial $O_3$ concentrations and the United States National Crop Loss Assessment Network (NCLAN) studies indicating that $O_3$ damage within crops occurs above this threshold (Heck et al., 1984; Lesser et al., 1990; Feng and Kobayashi, 2009). When the daily M7 $O_3$ concentration exceeds this threshold, photosynthesis is reduced by a factor between 0.0 to 1.0 (Eq. (1)) and leaf senescence is accelerated by a factor between 0.0 to 1.0 (Eq.

125      (5)).

Eq. (1) does not include the interaction of $O_3$ stress with water deficit stress or elevated atmospheric $CO_2$. To consider these combined interactions on crop growth ($PRFO_3$), $FO_3$ was modified using Eq. (2):

$$PRFO_3 = \min\left(1.0, \left(\frac{FO_3 * PCO_2}{SWFAC}\right)\right),$$      (2)

where $PCO_2$ is the atmospheric $CO_2$ effect on potential daily dry matter production and $SWFAC$ is the water stress

factor on photosynthesis (Jones and Kiniry, 1986; Ritchie et al., 1987; Jones et al., 2003). Since $PCO_2$ is always greater than one, multiplying by the $CO_2$ effect mitigates the reduction caused by $FO_3$. Because $SWFAC$ is a reduction factor between zero and one, dividing by this factor decreases the reduction from $FO_3$ under increased water deficit stress conditions.

The simulated daily biomass production (CARBO, g plant$^{-1}$ day$^{-1}$) within the models was calculated based on the

existing photosynthesis stress factors with the addition of $PRFO_3$ using Eq. (3) for maize and Eq. (4) for rice:

$$CARBO_{maize} = PCARB * \min(PRFT, SWFAC, NSTRES, PSTRES_1, KSTRES, PRFO_3) * SLPF,$$      (3)

$$CARBO_{rice} = PCARB * \min(PRFT, SWFAC, NSTRES, TSHOCK, PSTRES_1, KSTRES, PRFO_3) * SLPF,$$      (4)



where PCARB is daily potential dry matter production of the crop accounting for light interception, radiation use efficiency, and the $CO_2$ effect on photosynthesis (g plant[-1]), PRFT, SWFAC, NSTRES, TSHOCK (CERES-Rice only),

PSTRES$_1$, KSTRES, and PRFO$_3$ are the temperature, soil water, Nitrogen, transplanting shock, Phosphorous, Potassium, and $O_3$ stress factors on photosynthesis, respectively, and SLPF is the soil fertility factor (Jones and Kiniry, 1986; Ritchie et al., 1987; Jones et al., 2003).

Leaf senescence acceleration due to $O_3$ stress (SLFO$_3$) was added to the models using Eq. (5):

$$SLFO_3 = \max\left(0.0, -\left(\frac{SFOZ_1}{1000}\right) * OZON_7 + \left(1.0 + \left(\frac{SFOZ_1}{1000}\right) * 25.0\right)\right),$$   (5)

where SFOZ$_1$ is the $O_3$ stress parameter for leaf senescence calibrated for different $O_3$ sensitivities of cultivars divided by a decimal correction factor of 1000 (to ensure the SFOZ$_1$ parameter value ranges between 0.0 and 1.0 in the model ecotype file). The SLFO$_3$ factor was then included in the existing daily rate of leaf area senescence function (PLAS, cm$^2$ day[-1]) within the models as shown in Eq. (6) for maize and Eq. (7) for rice:

$$PLAS_{maize} = (PLA - SENLA) * \left(1 - \min(SLFW, SLFC, SLFT, SLFN, SLFP, SLFO_3)\right),$$   (6)

$$PLAS_{rice} = (PLA - SENLA) * \left(1 - \min(SLFW, SLFC, SLFT, SLFN, SLFP, SLFK, SLFO_3)\right),$$   (7)

where PLA is daily plant leaf area (cm$^2$ plant[-1]), SENLA is daily normal leaf senescence (cm$^2$ plant[-1]), and SLFW, SLFC, SLFT, SLFN, SLFP, SLFK, and SLFO$_3$ are the leaf senescence stress factors due to water, light competition, temperature, Nitrogen, Phosphorous, Potassium (CERES-Rice only), and $O_3$ stress, respectively (Jones and Kiniry,

1986; Ritchie et al., 1987; Jones et al., 2003).

### 2.2.2 CROPGRO-Soybean model

The effects of $O_3$ were incorporated into the CROPGRO-Soybean model using a similar approach as described in the CERES crop models. $O_3$ was added into the model using the same FO$_3$ and PRFO$_3$ factors as in Eq. (1) and Eq. (2) (for Eq. (2), PCO$_2$ is called PRATIO in CROPGRO-Soybean). However, CROPGRO-Soybean calculates daily

photosynthesis differently than the other models and has two different photosynthesis calculation options, leaf or canopy photosynthesis (Wilkerson et al., 1983; Boote and Pickering, 1994; Jones et al., 2003). This study focuses on the default leaf photosynthesis calculation option (which was modified to read in the $CO_2$ ratio effect for the PRFO$_3$ interaction). The daily gross photosynthesis (PG, g [CH$_2$O] m$^{-2}$ day[-1]) within the model was calculated based on the limiting photosynthesis stress factors using Eq. (8) for leaf photosynthesis and Eq. (9) for canopy photosynthesis:

$$PG_{leaf} = \left(\frac{PGDAY}{44.0} * 30.0 * SLPF\right) * \min(SWFAC, PRFO_3) * PSTRES_1,$$   (8)

$$PG_{canopy} = PTSMAX * SLPF * PG_{FAC} * TPG_{FAC} * E_{FAC} * PGSLW * PRATIO * PGLFMX * \min(SWFAC, PRFO_3),$$   (9)

where PGDAY is daily potential photosynthesis (g [CH$_2$O] m$^{-2}$ day[-1]), SWFAC, PSTRES$_1$, and PRFO$_3$ are the soil water, Phosphorous, and $O_3$ stress factors on photosynthesis, respectively. PTSMAX is the potential amount of CH$_2$O that can be produced for the full canopy (g [CH$_2$O] m$^{-2}$ day[-1]), PG$_{FAC}$ is a factor to compute daily PG as a function of





leaf area index, $TPG_{FAC}$ is a reduction factor for specific leaf area due to less optimal daytime temperature, $E_{FAC}$ is the effect of Nitrogen and Phosphorous stress on daily canopy photosynthesis, PGSLW is the relative effect of leaf thickness on daily canopy photosynthesis, and PRATIO is the relative effect of atmospheric $CO_2$ on daily canopy photosynthesis (Boote and Pickering, 1994).

Leaf senescence acceleration due to $O_3$ stress ($SLFO_3$) was added to CROPGRO-Soybean using Eq. (10):

$$SLFO_3 = \max\left(0.0, \left(\frac{SFOZ_1}{1000}\right) * OZON_7 - \left(\left(\frac{SFOZ_1}{1000}\right) * 25.0\right) * WTLF\right),\tag{10}$$

where WTLF is the dry mass of leaf tissue ($g_{leaf}$ m$^{-2}$). The CROPGRO leaf senescence routine is based on increasing WTLF, which is different from the CERES leaf senescence reduction factor, so $SLFO_3$ has the opposite trend when compared to the CERES model calculation (Fig. 1). The $SLFO_3$ factor was then included in the existing daily defoliation due to daily leaf senescence (SLDOT, g m$^{-2}$ day$^{-1}$) calculation within the model as shown in Eq. (11):

$$SLDOT = SLDOT_n + \max(SLNDOT, SLFO_3),\tag{11}$$

where $SLDOT_n$ is the natural daily leaf senescence and SLNDOT and $SLFO_3$ are the daily leaf senescence due to water and $O_3$ stress (g m$^{-2}$ day$^{-1}$), respectively.

### 2.2.3 DSSAT-NWheat model

The incorporation of $O_3$ into the NWheat crop model was described and validated in Guarin et al., (2019) and was
used as the reference for the maize, rice, and soybean models. The approach used the same $FO_3$ and $PRFO_3$ equations as in Eq. (1) and (2) (note that the NWheat equations were simplified from Guarin et al., (2019) by the decimal correction factor and single $FOZ_1$ parameter as in Eq. (1) for consistency among all models) and a similar $SLFO_3$ shown in Eq. (12):

$$SLFO_3 = \left(\frac{SFOZ_1}{10}\right) * OZON_7 + \left(1.0 - \left(\frac{SFOZ_1}{10} * 25.0\right)\right).\tag{12}$$

The $O_3$ effect for the different cultivar sensitivities is controlled by the $FOZ_1$ and $SFOZ_1$ parameters, as in the other models (the $SFOZ_1$ parameter is divided by 10 to ensure that the value ranges between 0.0 and 1.0 in the model ecotype file). The decimal correction factors vary between the crop models because the different models calculate stresses using different magnitudes.

The $FO_3$ and $SLFO_3$ responses calculated over increasing M7 $O_3$ concentrations are illustrated for each model in Fig.
1 using the parameter values for different $O_3$ cultivar classifications shown in Table 1. The $FOZ_1$ and $SFOZ_1$ parameter values for all models were determined from the cultivar sensitivities observed in the field experiments (section 2.3) and the sensitivities derived from the $O_3$ exposure relationships from the literature (section 2.5).




**Figure 1: Functions for the O₃ photosynthesis reduction factor without interaction of water deficit stress and CO₂ fertilization effect (FO₃) (first column) and the O₃ leaf senescence acceleration stress factor (SLFO₃) (second column) under increasing mean 7-hour (M7) O₃ concentrations for the (a, b) CERES-Maize, (c, d) CERES-Rice, (e, f) CROPGRO-Soybean, and (g, h) NWheat models. Each figure shows three different O₃ sensitivity cultivar classifications derived from the O₃ exposure-yield responses from the literature: tolerant (blue solid line), intermediate (gold short-dash line), and sensitive (magenta long-dash line). SLFO₃ for CROPGRO-Soybean (Eq. (10)) shown with leaf tissue dry mass (WTLF) of 1 g m⁻² for simplicity. Steeper slopes indicate a higher sensitivity to O₃ for both FO₃ and SLFO₃. Table 1 shows the parameters used in the equations for each classification of O₃ sensitivity (Eq. (1), (5), (10), and (12)).**




**Table 1: Summary of the O$_3$ photosynthesis stress parameters (FOZ$_1$) and the O$_3$ leaf senescence stress parameters (SFOZ$_1$) used in the FO$_3$ and SLFO$_3$ calculations (Eq. (1), (5), (10), and (12)) for the four DSSAT models under three different O$_3$ sensitivity cultivar classifications. The CERES and CROPGRO parameter values were determined from the O$_3$ exposure-yield responses in the literature (Fig. S2, Fig. S3). NWheat parameter values were from Guarin et al., (2019) and confirmed with the literature.**

| O$_3$ sensitivity cultivar classifications | CERES-Maize | | CERES-Rice | | CROPGRO-Soybean | | NWheat | |
|---|---|---|---|---|---|---|---|---|
| | FOZ$_1$ | SFOZ$_1$ | FOZ$_1$ | SFOZ$_1$ | FOZ$_1$ | SFOZ$_1$ | FOZ$_1$ | SFOZ$_1$ |
| Tolerant | 0.15 | 0.10 | 0.10 | 0.08 | 0.15 | 0.15 | 0.06 | 0.08 |
| Intermediate | 0.30 | 0.20 | 0.30 | 0.10 | 0.25 | 0.25 | 0.10 | 0.25 |
| Sensitive | 0.60 | 0.40 | 0.65 | 0.12 | 0.40 | 0.35 | 0.50 | 0.40 |

**2.3 Observed O$_3$ exposure field experiments**

In general, detailed field experiments of crop growth under elevated O$_3$ conditions for different crops are scarce and limit the granularity of model calibration. All field experiments examined in this study used dominant management conditions to limit other stresses besides O$_3$, e.g., water deficit or N stress, so the simulations assumed negligible outside stresses.

For maize, the FACE experiment conducted at Champaign, Illinois, USA (40.03 °N, 88.27 °W, 230 m elevation) in 2018 was used for calibrating the CERES-Maize model (Choquette et al., 2020). The maize FACE experiment consisted of six cultivars grown under an ambient and an elevated O$_3$ treatment with $n = 4$ (Table 2). Since there was only one year of data, the model was validated against the O$_3$ exposure-relative yield response functions from the literature (section 2.5). The daily maximum temperature (TMAX), minimum temperature (TMIN), and precipitation

(RAIN) weather data were collected from the nearby National Oceanic and Atmospheric Administration (NOAA) Willard airport weather station and the daily incoming solar radiation (SRAD) was collected from the National Aeronautics and Space Administration (NASA) Prediction Of Worldwide Energy Resources (POWER) database (https://power.larc.nasa.gov/). The soil consisted of the Drummer silty clay loam soil series, and the soil parameters for this series were obtained from the United States Department of Agriculture (USDA) Natural Resources

Conservation Service (NRCS) Web Soil Survey database (Table S1) (Nrcs, 2023). The cultivars were planted in two 3.5 m rows with a row spacing of 0.76 m on May 13, 2018 (Choquette et al., 2020). The hourly O$_3$ fumigation (from 10:00 to 18:00) began on May 25, 2018 and ended on August 14, 2018 and was used to calculate the daily M7 O$_3$ concentrations. The cultivar plots were harvested at maturity on September 21, 2018. N and water deficit stress were reported to be non-limiting, so the simulations used the non-limiting N setting within the model and the simulated

water stress was confirmed to be non-limiting with the provided rainfall. The DSSAT cultivar parameters were calibrated for phenology and growth under negligible stress conditions using the treatment with the ambient O$_3$ concentration (38 ppb) for each cultivar. After the phenology and growth cultivar parameters were calibrated, the FOZ$_1$ and SFOZ$_1$ O$_3$ response parameters were calibrated using the yield response from the elevated O$_3$ concentration treatments (Fig. S1 (a)).

For soybean, data from the FACE experiment conducted at the same location in Champaign, Illinois, USA (40.03 °N, 88.27 °W, 230 m elevation) in 2009 and 2010 was used for model testing (Betzelberger et al., 2012). The 2009 data was used for model calibration and the 2010 data was used for model validation. These data were previously used to



incorporate $O_3$ effects on leaf photosynthesis into the JULES model (Leung et al., 2020). The SoyFACE experiment consisted of seven soybean cultivars grown under nine $O_3$ treatments with different target concentrations (Table 2). The hourly $O_3$ fumigation data (plots fumigated for 8 to 9 hours daily except when leaves were wet) for each treatment was recorded in situ and was used to calculate the daily M7 $O_3$ concentrations (Betzelberger et al., 2012). The weather data was collected from the same sources as used in the maize experiment (NOAA and NASA POWER), and the soil consisted of either the Drummer silty clay loam or the Flanagan silt loam series which were obtained from the USDA NRCS Web Soil Survey database (Table S1). The initial soil conditions of the simulations were set at 95% available water content and 100 kg N ha$^{-1}$ to minimize water and N stress. The cultivars were planted in plots eight rows wide and 5.4 m long, with a row spacing of 0.38 m, on June 9, 2009 and May 27, 2010. The $O_3$ fumigation started on June 29, 2009 and June 6, 2010, and ended on September 27, 2009 and September 17, 2010. The cultivar plots were harvested at maturity on October 20, 2009 and September 30, 2010. For each specified cultivar maturity group (Betzelberger et al., 2012), the corresponding default DSSAT maturity group parameters were used as reference and then calibrated for phenology and growth under negligible stress using the treatment with the ambient $O_3$ concentration (37 ppb). After the phenology and growth cultivar parameters were calibrated, the $FOZ_1$ and $SFOZ_1$ $O_3$ response parameters were calibrated using the yield response from the elevated $O_3$ concentration treatments (Fig. S1 (b)). The parameters for both maize and soybean were calibrated using the using the one-factor-at-at-time method (Morris, 1991) until the best fit was found for the phenology, growth, and relative yield loss for each cultivar across all $O_3$ treatments.

**Table 2: $O_3$ fumigation target concentration and average mean 7-hour (M7, 9:00 – 15:59 hr) $O_3$ concentrations for the 2018 maize FACE experiment (Choquette et al., 2020) and the 2009 and 2010 soybean SoyFACE experiments (Betzelberger et al., 2012).**

| Crop experiment | $O_3$ fumigation target concentration (ppb) | Average M7 $O_3$ concentration (ppb) |
|---|---|---|
| Maize 2018 | Ambient | 38 |
|  | 100 | 77 |
| Soybean 2009 | Ambient | 37 |
|  | 40 | 39 |
|  | 55 | 47 |
|  | 70 | 57 |
|  | 85 | 61 |
|  | 110 | 75 |
|  | 130 | 96 |
|  | 160 | 102 |
|  | 200 | 126 |
| Soybean 2010 | Ambient | 37 |
|  | 55 | 46 |
|  | 70 | 52 |
|  | 85 | 59 |
|  | 110 | 69 |
|  | 130 | 76 |
|  | 150 | 70 |
|  | 170 | 84 |
|  | 190 | 84 |






For rice, there was no $O_3$ field experiment data readily available, thus a representative rice-producing location in the main North American rice-producing area at Stuttgart, Arkansas, USA (34.50 °N, 91.55 °W, 60 m elevation) (Usda Nass, 2010) was simulated with the default DSSAT North American rice cultivar. 2009 was selected for consistency with the soybean simulations. The weather data was collected from the NASA POWER database and the dominant

soil series for Arkansas County, Dewitt silt loam, was determined from the USDA NRCS Web Soil Survey database (Table S1) (Nrcs, 2023). The initial soil conditions of the simulations were set at 100% available water content and 100 kg N ha$^{-1}$ to ensure negligible water and N stress. Four 50 kg N ha$^{-1}$ fertilizer applications were applied throughout the season to ensure negligible N stress in the simulations. The cultivar was planted on April 20, 2009 based on the most active planting dates recorded for Arkansas in the USDA Field Crops handbook (Usda Nass, 2010), and the

harvest date was automatically calculated based on when the model simulations reached physiological maturity. The default DSSAT North American rice cultivar parameters were used, and the $FOZ_1$ and $SFOZ_1$ $O_3$ response parameters were calibrated using the yield response from the elevated $O_3$ exposure functions from the literature (section 2.5).

For wheat, the NWheat model was calibrated and validated using an air exclusion system $O_3$ exposure wheat field experiment conducted in Wake County, North Carolina, USA (35.73 °N, 78.68 °W, 116 m elevation) and is described

in detail in Guarin et al., (2019).

**2.4 Sensitivity analysis of $O_3$ equations and parameters**

A sensitivity analysis for maize, rice, and soybean was conducted using simulations of nine constant daily M7 $O_3$ concentrations of 25, 40, 50, 60, 70, 80, 90, 100, and 120 ppb with different $FOZ_1$ and $SFOZ_1$ parameter values under combinations between normal or 50% reduced rainfall and 350 ppm or 550 ppm $CO_2$ concentrations to confirm that

the $O_3$ modifications and stress interactions within the models were behaving as expected. The simulated locations and management setup for each crop were the same as the field experiments described above (section 2.3). For wheat, the sensitivity analysis was based on the 1993 FACE experiment conducted in Maricopa, Arizona (33.06 °N, 111.98 °W, 361 m elevation) (Hunsaker et al., 1996; Kimball et al., 1999; Kimball et al., 2017). The simulation setup for the Maricopa FACE experiment used the same 9 M7 $O_3$ concentrations with either a "Wet" irrigation schedule (total of

629 mm sub-surface drip irrigation at 0.23 m from planting to harvest) or a "Dry" irrigation schedule (total of 347 mm sub-surface drip irrigation at 0.23 m from planting to harvest) under 350 ppm and 550 ppm $CO_2$ concentrations to examine the $O_3$-$CO_2$-water interactions as detailed in Guarin et al., (2019). For all crops, each $O_3$ parameter was first tested independently to examine the individual effects on photosynthesis and leaf senescence, i.e., when examining $FOZ_1$, $SFOZ_1$ was set to zero and vice versa.

**2.5 Observed $O_3$ exposure relationships based on the literature**

To confirm that the models were able to reproduce the observed relative yield loss due to $O_3$ stress, the simulated results were compared to well-known literature reports of $O_3$ exposure metrics and yield response for each crop using the M7 $O_3$ concentrations. The simulated locations and management conditions were the same experimental conditions as described above for each crop. For each crop, different $O_3$ classification of cultivar sensitivities were defined based



on more severe response to O₃ stress, i.e., tolerant, intermediate, and sensitive. These classifications of cultivar O₃ sensitivity were determined using the extensive literature review data from Mills et al. (2018a) combined with the maize and soybean FACE data for a total of 9 maize cultivars, 50 rice cultivars, 49 soybean cultivars, and 23 wheat cultivars. The literature review consisted of O₃ exposure experiments conducted in open-top chambers, experimental fields, or greenhouses and included the experiments that contributed to the widely applied Weibull O₃ response

function (Heck et al., 1984; Adams et al., 1989; Lesser et al., 1990; Wang and Mauzerall, 2004; Tai et al., 2021; Feng et al., 2022). The selection criteria of the data are described in detail in Mills et al. (2018a).

The yield data from the literature experiments were standardized as performed by Mills et al. (2018a) and described by Osborne et al. (2016). For each experiment, linear regression was used to determine the yield at 25 ppb M7 O₃ and this value was the reference for calculating the relative yield, i.e., relative yield was calculated as the actual observed

yield divided by the yield at 25 ppb O₃. The 25 ppb M7 O₃ threshold was chosen for proper comparison to the model results. After calculating the yield relative to 25 ppb M7 O₃, a linear regression for each cultivar was performed using R statistical software, v4.3.0, (R Core Team, 2023; Wickham, 2016; Wickham et al., 2023) to determine the O₃ exposure response (Fig. S2). The cultivar O₃ exposure responses were then classified into three evenly distributed quantiles, 0%-33%, 33%-66%, and 66%-100%, chosen to represent the three O₃ sensitivity classifications: sensitive,

intermediate, and tolerant, respectively (Fig. S3). These data were used to determine the model $FOZ_1$ and $SFOZ_1$ values of each of the O₃ cultivar classifications shown in Table 1 to evaluate if the models could accurately reproduce the O₃ exposure-yield responses.

## 3 Results

### 3.1 Calibration of crop models and simulated relative yield loss against O₃ exposure field experiments

The simulated phenology (anthesis [flowering] and physiological maturity dates), biomass, yield, and relative yield loss due to O₃ stress from the maize and soybean experiments were compared to the field observations to determine performance of the O₃ equations within the models (Tables 3 – 5, Fig. 2 and 3, Fig. S1). The relative yield loss due to O₃ stress was calculated by dividing the yield of each corresponding O₃ treatment over the control yield, i.e., the baseline O₃ treatment. There was no O₃ field experiment data for rice, so the rice O₃ parameter values and performance

were compared to the O₃ exposure-yield response functions from the literature (section 3.3).

The maize and soybean cultivars had different sensitivities to O₃ stress which were accounted for by using different $FOZ_1$ and $SFOZ_1$ values (Fig. S1). The calibrated CERES-Maize and CROPGRO-Soybean models simulated the physiological maturity within four days of the observations (Table 5; Root Mean Square Error (RMSE) = 0.0 days for maize 2018, 3.70 days for soybean 2009, and 3.30 days for soybean 2010). The calibrated CERES-Maize model was

able to reproduce the yield and relative yield loss very well across all six cultivars (Fig. 2; RMSE = 107 kg ha⁻¹ and 2%; r² = 0.99 and 0.99, respectively). This ideal model performance was because only two O₃ treatments were available for each maize cultivar which simplified the calibration process (Fig. S1 (a)). The CROPGRO-Soybean model was able to reproduce the biomass, yield, and relative yield loss due to O₃ stress well for the calibration year, 2009 (Fig. 3 (a), (b), (c); RMSE = 1179 kg ha⁻¹, 328 kg ha⁻¹, and 10%; r² = 0.81, 0.88, and 0.85), and acceptably for




the evaluation year, 2010, across all seven cultivars (Fig. 3 (d), (e), (f); RMSE = 3339 kg ha$^{-1}$, 1291 kg ha$^{-1}$, and 16%; $r^2$ = 0.59, 0.71, and 0.66). The model overestimated biomass and yield for all cultivars and treatments in 2010, which was likely the result of a factor outside of the model setup that mitigated the increased incoming solar radiation when compared to 2009 (section 4.3). The calibration and evaluation for the NWheat model was conducted and validated in Guarin et al. (2019), where the model reproduced the observed relative yield due to O$_3$ stress with a Normalized

Root Mean Square Error (NRMSE) of 23% and an $r^2$ of 0.94, 0.91, and 0.88 for the tolerant, intermediate, and sensitive O$_3$ sensitive cultivar classifications.

**Table 3: CERES-Maize cultivar and O$_3$ parameters used to simulate the six maize cultivars from the 2018 FACE field experiment (Choquette et al., 2020). P1 = Thermal time from seedling emergence to the end of the juvenile phase**
**(expressed in degree days above a base temperature of 8 °C), P2 = Extent to which daily development is delayed for each hour increase in photoperiod above the longest photoperiod at which development proceeds at a maximum rate (which is considered to be 12.5 hours), P5 = Thermal time from silking to physiological maturity (expressed in degree days above a base temperature of 8 °C), G2 = Maximum possible number of kernels per plant, G3 = Kernel filling rate during the linear grain filling stage and under optimum conditions (mg day$^{-1}$), PHINT = Phylochron interval, i.e., the interval in**
**thermal time (degree days) between successive leaf tip appearances, FOZ$_1$ = O$_3$ effect on photosynthesis, and SFOZ$_1$ = O$_3$ effect on leaf senescence.**

| Cultivar | P1 | P2 | P5 | G2 | G3 | PHINT | FOZ$_1$ | SFOZ$_1$ |
|---|---|---|---|---|---|---|---|---|
| B73 x Hp301 | 110 | 0.5 | 700 | 700 | 8.5 | 38.9 | 0.40 | 0.20 |
| B73_x_Mo17 | 110 | 0.5 | 700 | 700 | 5.9 | 38.9 | 0.20 | 0.15 |
| B73_x_NC338 | 110 | 0.5 | 700 | 700 | 7.8 | 38.9 | 0.65 | 0.40 |
| Mo17 x Hp301 | 110 | 0.5 | 700 | 700 | 5.5 | 38.9 | 0.10 | 0.10 |
| Mo17_x_NC338 | 110 | 0.5 | 700 | 700 | 8.5 | 38.9 | 0.50 | 0.30 |
| NC338_x_Hp301 | 110 | 0.5 | 700 | 700 | 5.1 | 38.9 | 0.10 | 0.10 |





**Table 4:** CROPGRO-Soybean cultivar and $O_3$ parameters used to simulate the seven soybean cultivars based on the maturity groups defined in the SoyFACE field experiment (Betzelberger et al., 2012). CSDL = Critical Short Day Length below which reproductive development progresses with time (positive for shortday plants) (per hour), EM-FL = Time between plant emergence and flower appearance (R1) (photothermal days), FL-SH = Time between first flower and first pod (R3) (photothermal days), FL-SD = Time between first flower and first seed (R5) (photothermal days), SD-PM = Time between first seed (R5) and physiological maturity (R7) (photothermal days), FL-LF = Time between first flower (R1) and end of leaf expansion (photothermal days), LFMAX = Maximum leaf photosynthesis rate at 30 °C, 350 vpm $CO_2$, and high light (mg $CO_2$ m$^{-2}$ s$^{-1}$), SLAVR = Specific leaf area of cultivar under standard growth conditions (cm$^2$ g$^{-1}$), SIZLF = Maximum size of full leaf (three leaflets) (cm$^2$), XFRT = Maximum fraction of daily growth that is partitioned to seed and shell, WTPSD = Maximum weight per seed (g), SFDUR = Seed filling duration for pod cohort at standard growth conditions (photothermal days), SDPDV = Average seed per pod under standard growing conditions (number per pod), PODUR = Time required for cultivar to reach final pod load under optimal conditions (photothermal days), THRSH = Threshing percentage. The maximum ratio of (seed per (seed + shell)) at maturity, SDPRO = Fraction protein in seeds (g(protein) per g(seed)), SDLIP = Fraction oil in seeds (g(oil) per g(seed)), $FOZ_1$ = $O_3$ effect on photosynthesis, and $SFOZ_1$ = $O_3$ effect on leaf senescence.

| Cultivar | Maturity Group | CSDL | PPSEN | EM-FL | FL-SH | FL-SD | SD-PM | FL-LF | LFMAX | SLAVR | SIZLF | XFRT | WTPSD | SFDUR | SDPDV | PODUR | THRSH | SDPRO | SDLIP | FOZ$_1$ | SFOZ$_1$ |
|---|---|---|---|---|---|---|---|---|---|---|---|---|---|---|---|---|---|---|---|---|---|
| Pioneer 93B15 | 3 | 13.1 | 0.285 | 19.0 | 6 | 14.0 | 33.2 | 26 | 1.20 | 375 | 180 | 1 | 0.16 | 23 | 2.2 | 10 | 77 | 0.405 | 0.205 | 0.25 | 0.25 |
| Dwight | 2 | 12.9 | 0.249 | 17.4 | 6 | 13.5 | 32.4 | 26 | 1.00 | 375 | 180 | 1 | 0.16 | 23 | 2.2 | 10 | 77 | 0.405 | 0.205 | 0.20 | 0.20 |
| HS93-4118 | 4 | 13.3 | 0.294 | 19.4 | 7 | 15.0 | 34.0 | 26 | 1.05 | 375 | 180 | 1 | 0.16 | 23 | 2.2 | 10 | 77 | 0.405 | 0.205 | 0.30 | 0.30 |
| IA-3010 | 3 | 13.2 | 0.285 | 19.0 | 6 | 14.0 | 33.2 | 26 | 1.01 | 375 | 180 | 1 | 0.16 | 23 | 2.2 | 10 | 77 | 0.405 | 0.205 | 0.25 | 0.25 |
| LN97-15076 | 4 | 13.2 | 0.294 | 19.4 | 7 | 15.0 | 34.0 | 26 | 1.08 | 375 | 180 | 1 | 0.19 | 23 | 2.2 | 10 | 77 | 0.405 | 0.205 | 0.30 | 0.30 |
| Loda | 2 | 12.7 | 0.249 | 17.4 | 6 | 13.5 | 32.4 | 26 | 1.03 | 375 | 180 | 1 | 0.19 | 23 | 2.2 | 10 | 77 | 0.405 | 0.205 | 0.30 | 0.30 |
| Pana | 3 | 13.0 | 0.285 | 19.0 | 6 | 14.0 | 33.2 | 26 | 1.00 | 375 | 180 | 1 | 0.15 | 23 | 2.2 | 10 | 77 | 0.405 | 0.205 | 0.25 | 0.30 |





**Table 5: Observed and simulated anthesis day and maturity day for the six maize cultivars from the 2018 FACE experiment (Choquette et al., 2020) and the seven soybean cultivars from the 2009 and 2010 soybean SoyFACE experiments (Betzelberger et al., 2012). The observed maturity dates were estimated from the single reported harvest date for all cultivars but there may have been minor variation between the different cultivars. Observed anthesis was not available for soybean.**

| Crop experiment | Cultivar | Observed anthesis (dap) | Simulated anthesis (dap) | Observed maturity (dap) | Simulated maturity (dap) |
|---|---|---|---|---|---|
| Maize 2018 | B73 x Hp301 | 48 | 48 | 97 | 97 |
| | B73 x Mo17 | 48 | 48 | 97 | 97 |
| | B73_x_NC338 | 48 | 48 | 97 | 97 |
| | Mo17 x Hp301 | 48 | 48 | 97 | 97 |
| | Mo17 x NC338 | 48 | 48 | 97 | 97 |
| | NC338 x Hp301 | 48 | 48 | 97 | 97 |
| Soybean 2009 | Pioneer93B15 | | 52 | 133 | 131 |
| | Dwight | | 48 | 133 | 126 |
| | HS93-4118 | | 53 | 133 | 133 |
| | IA-3010 | | 50 | 133 | 128 |
| | LN97-15076 | | 55 | 133 | 137 |
| | Loda | | 52 | 133 | 132 |
| | Pana | | 54 | 133 | 134 |
| Soybean 2010 | Pioneer93B15 | | 48 | 126 | 129 |
| | Dwight | | 44 | 126 | 125 |
| | HS93-4118 | | 48 | 126 | 129 |
| | IA-3010 | | 47 | 126 | 126 |
| | LN97-15076 | | 50 | 126 | 131 |
| | Loda | | 48 | 126 | 130 |
| | Pana | | 51 | 126 | 131 |


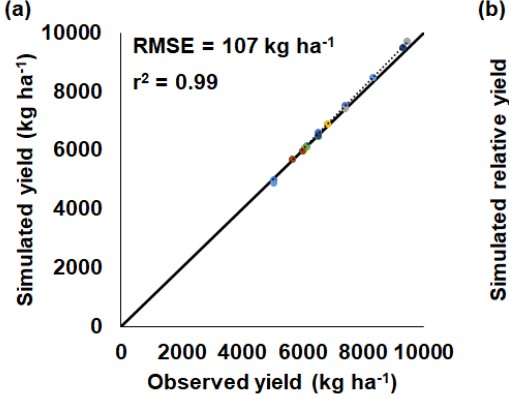

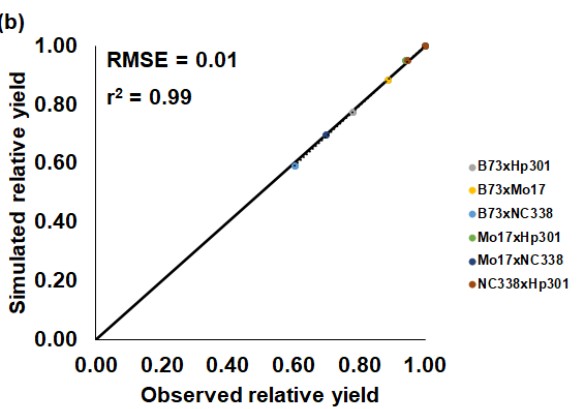

**Figure 2: CERES-Maize model calibration of the 2018 FACE O₃ field experiment conducted in Champaign, Illinois, USA (Choquette et al., 2020). Simulated and observed (a) yield and (b) relative yield loss due to elevated O₃ stress (compared to the ambient control treatment) for six maize cultivars (colored points). The root-mean-square error (RMSE) and coefficient of determination (r²) show the model performance across all cultivars. Solid black line shows 1:1 comparison and dotted black line shows linear fit across all cultivars. For maize only one year of experimental data was available for calibration and evaluation. The model cultivar parameters are shown in Table 3.**



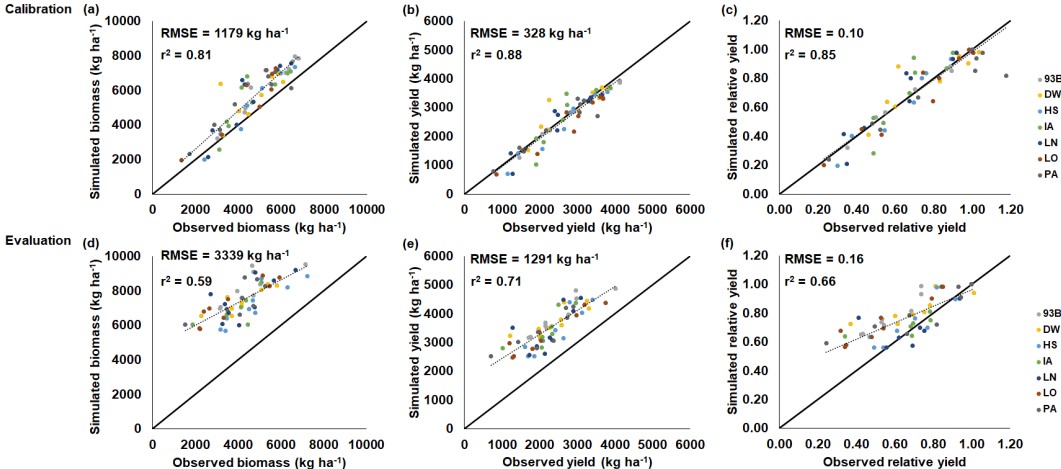

**Figure 3: CROPGRO-Soybean model performance and evaluation of the SoyFACE O₃ field experiment conducted in Champaign, Illinois, USA (Betzelberger et al., 2012). Simulated and observed (a, d) above-ground biomass, (b, e) yield, and (c, f) relative yield loss in response to the nine progressive O₃ increasing treatments (Table 2) for seven soybean cultivars (colored points). Relative yield loss is compared to the ambient control treatment within each year. The 2009 SoyFACE field experiment was used for model calibration (a, b, c), and the 2010 SoyFACE field experiment was used for model evaluation (d, e, f). The root-mean-square error (RMSE) and coefficient of determination (r²) show the model performance across all cultivars. Solid black line shows 1:1 comparison and dotted black line shows linear fit across all cultivars. The model cultivar parameters are shown in Table 4.**

### 3.2 Sensitivity analysis and combined effects of O₃, CO₂, and water deficit stress on yields

The simulated relative yield losses due to O₃ stress increased for all crops as the M7 O₃ concentrations increased above the 25 ppb threshold when examining the photosynthesis and leaf senescence responses independently, as expected (Figs. 4 – 7). The simulated actual yields for all crops are shown in the Supplementary Tables S2 – S9. Wheat was the most sensitive crop to O₃ stress of the four crops examined (compare slopes in Figs. 4 – 7 (a) and (b)) which agrees with previous literature (Mills et al., 2018a). For each model, simulations using a FOZ₁ or SFOZ₁ example value of 0.5 were examined in more detail to illustrate the O₃-CO₂-water interactions (Figs. 4 – 7 (c) and (d), respectively). For all crops, the Dry/reduced rainfall and low CO₂ treatment produced the lowest yields while the Wet/normal rainfall and high CO₂ produced the highest yields (Tables S2 – S9). The simulated O₃ effect was larger when water deficit stress was non-limiting, i.e., the higher rainfall and irrigated treatments experienced larger losses due to O₃ stress because of increased stomatal uptake. The simulated O₃ effect was reduced under the higher CO₂ concentrations, thus capturing the responses from stomatal closure and the photosynthetic benefits from the CO₂ fertilization effect.



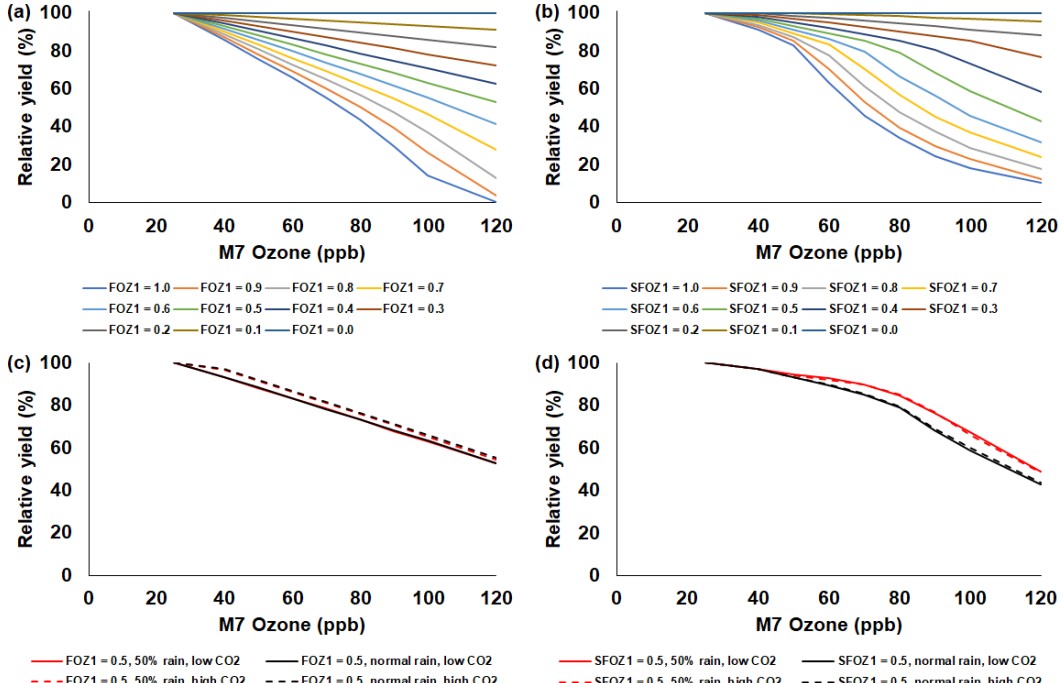

**Figure 4: Sensitivity analysis using the CERES-Maize model to simulate $O_3$ relative yield loss for a range of (a) the photosynthesis $O_3$ stress parameter ($FOZ_1$) and (b) the leaf senescence $O_3$ stress parameter ($SFOZ_1$) values under the normal rainfall and 350 ppm $CO_2$ scenario, and an example of (c) $FOZ_1$ and (d) $SFOZ_1$ set at 0.5 under the 50% reduced rainfall and 350 ppm $CO_2$ (solid red line), normal rainfall and 350 ppm $CO_2$ (solid black line), 50% less rainfall and 550 ppm $CO_2$ (dashed red line), and normal rainfall and 550 $CO_2$ (dashed black line) scenarios. The Champaign, Illinois, USA**

**2018 FACE weather, soil, and dominant management conditions were used for the reference location. Each $O_3$ parameter was tested independently, i.e., when examining $FOZ_1$, $SFOZ_1$ was set to zero and vice versa. The simulated actual yields are shown in Tables S2 and S3.**



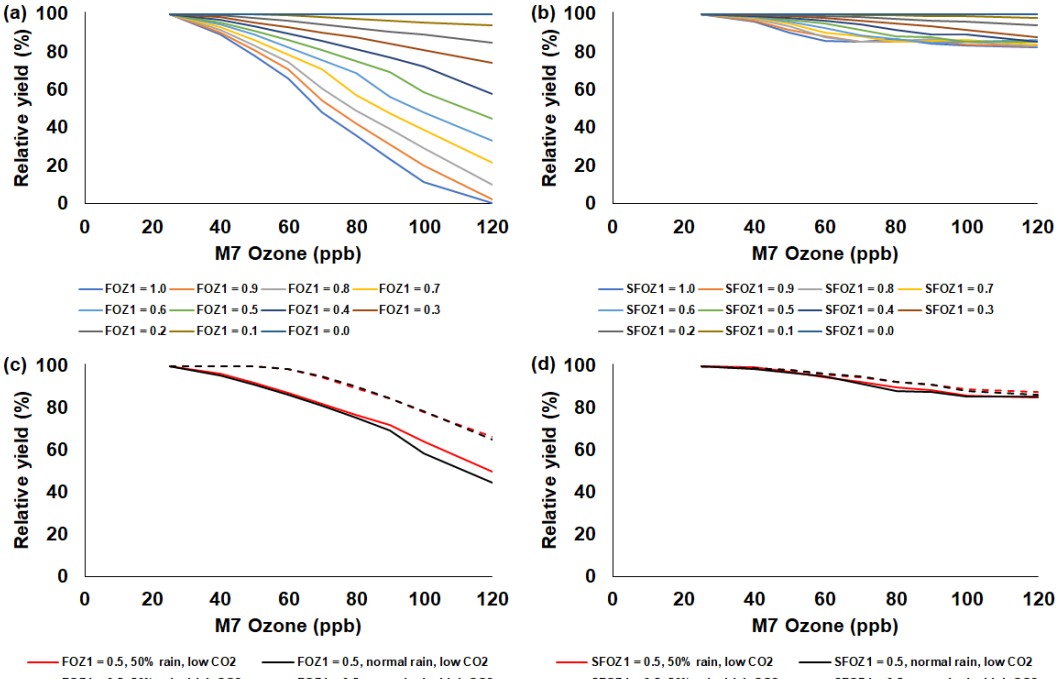

**Figure 5: Sensitivity analysis using the CERES-Rice model to simulate $O_3$ relative yield loss for a range of (a) $FOZ_1$ and (b) $SFOZ_1$ values under the normal rainfall and 350 ppm $CO_2$ scenario, and an example of (c) $FOZ_1$ and (d) $SFOZ_1$ set at 0.5 under the 50% reduced rainfall and 350 ppm $CO_2$ (solid red line), normal rainfall and 350 ppm $CO_2$ (solid black line), 50% less rainfall and 550 ppm $CO_2$ (dashed red line), and normal rainfall and 550 $CO_2$ (dashed black line) scenarios. The Stuttgart, Arkansas, USA 2009 weather, soil, and dominant management conditions were used for the reference location. Each $O_3$ parameter was tested independently, i.e., when examining $FOZ_1$, $SFOZ_1$ was set to zero and vice versa. The simulated actual yields are shown in Tables S4 and S5.**



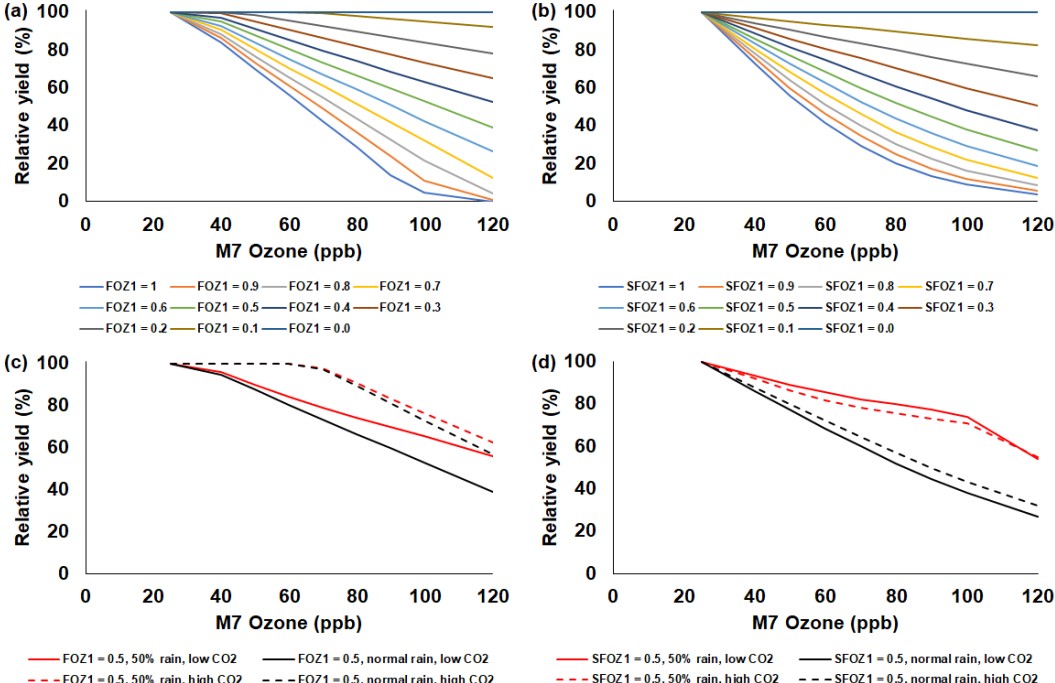

**Figure 6: Sensitivity analysis using the CROPGRO-Soybean model to simulate $O_3$ relative yield loss for a range of (a) $FOZ_1$ and (b) $SFOZ_1$ values under the normal rainfall and 350 ppm $CO_2$ scenario, and an example of (c) $FOZ_1$ and (d) $SFOZ_1$ set at 0.5 under the 50% reduced rainfall and 350 ppm $CO_2$ (solid red line), normal rainfall and 350 ppm $CO_2$ (solid black line), 50% less rainfall and 550 ppm $CO_2$ (dashed red line), and normal rainfall and 550 $CO_2$ (dashed black line) scenarios. The Champaign, Illinois, USA 2009 SoyFACE weather, soil, and dominant management conditions were used for the reference location. Each $O_3$ parameter was tested independently, i.e., when examining $FOZ_1$, $SFOZ_1$ was set to zero and vice versa. The simulated actual yields are shown in Tables S6 and S7. Figure S6 shows the relative biomass loss corresponding to $SFOZ_1$ (d) to explain the inverted $CO_2$ effect under the 50% rainfall treatment.**






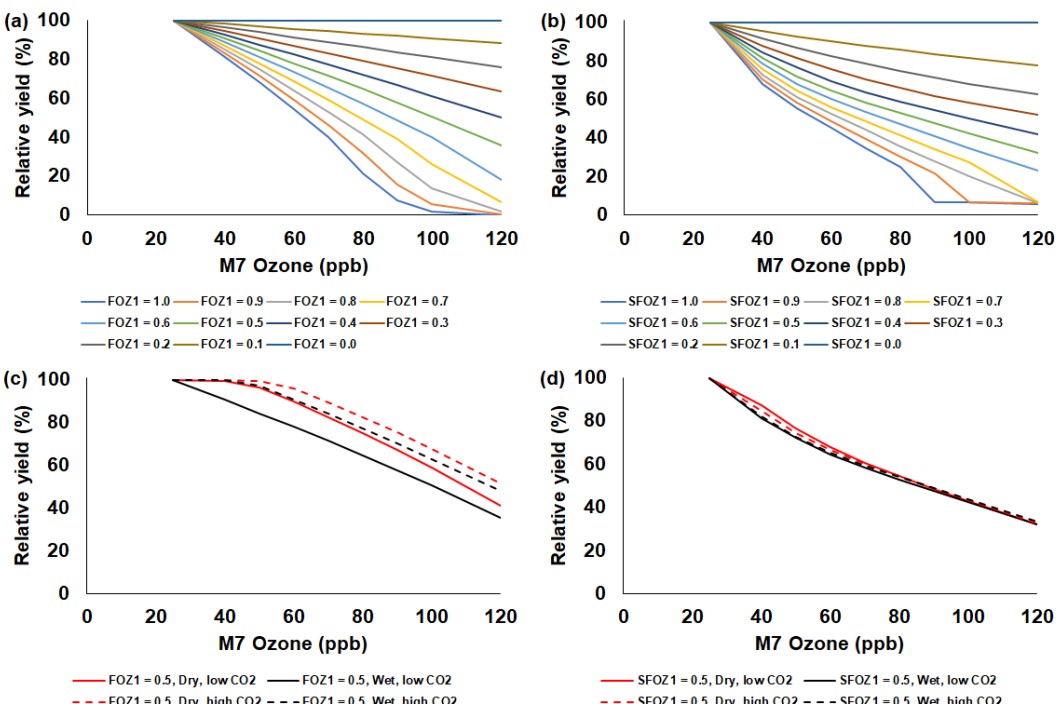


**Figure 7: Sensitivity analysis using the NWheat model to simulate O₃ relative yield loss for a range of (a) FOZ₁ and (b) SFOZ₁ values under the "Wet" irrigation and 350 ppm CO₂ scenario, and an example of (c) FOZ₁ and (d) SFOZ₁ set at 0.5 under the "Dry" irrigation and 350 ppm CO₂ (solid red line), "Wet" irrigation and 350 ppm CO₂ (solid black line), "Dry" irrigation and 550 ppm CO₂ (dashed red line), and "Wet" irrigation and 550 CO₂ (dashed black line) scenarios.**

**The Maricopa, Arizona, USA 1993 FACE weather, soil, and management conditions were used for the reference location (Kimball et al., 1999; Guarin et al., 2019). Each O₃ parameter was tested independently, i.e., when examining FOZ₁, SFOZ₁ was set to zero and vice versa. The simulated actual yields are shown in Tables S8 and S9.**

### 3.3 Simulated relative yield loss compared to O₃ relationships in the literature

For all crops, the literature showed a large range of relative yield losses due to O₃ stress caused by different cultivar O₃ sensitivities (Fig. S2). Wheat was the most sensitive crop to O₃ stress with an average yield loss of $0.70\% \pm 0.39$ (mean $\pm$ SD) per ppb M7 O₃ increase above 25 ppb, followed by soybean, maize, and then rice (average yield losses of $0.60\% \pm 0.39$, $0.39\% \pm 0.26$, and $0.32\% \pm 0.37$ per ppb M7 O₃ increase above 25 ppb, respectively) (average of slopes in Table S10). To encompass the high variability of yield losses, the cultivars were classified into the O₃

tolerant, intermediate, and sensitive cultivar O₃ sensitivities (Fig. S3). Since the cultivar sensitivities were not originally specified in the literature, the FOZ₁ and SFOZ₁ parameters used in the models were adjusted to provide the best fit across the O₃ exposure responses (Table 1). Overall, the models reproduced the simulated O₃ exposure relationships from the literature well; the RMSE for maize, rice, soybean, and wheat across all three O₃ exposure sensitivities were 6.6%, 7.8%, 4.0%, and 5.4%, respectively (Fig. 8). The models performed better (lower RMSE) for

the O₃ tolerant and O₃ intermediate cultivar sensitivities compared to the O₃ sensitive cultivar sensitivity, but all




models explained the variance well ($r^2 > 0.96$ across all $O_3$ sensitivities). This suggests that different combinations of $FOZ_1$ and $SFOZ_1$ can be calibrated for specific observations to emulate the variation in different $O_3$ exposure responses.

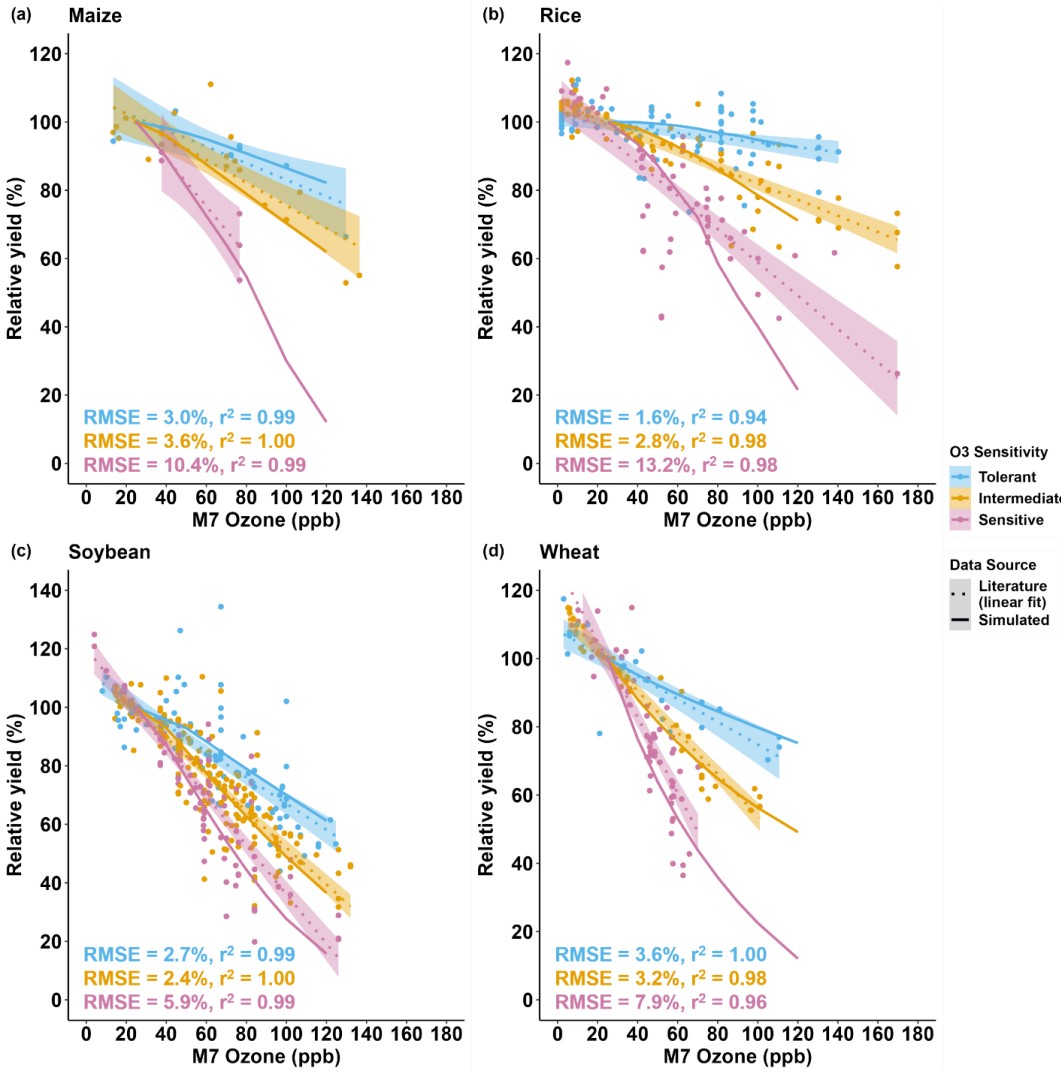


**Figure 8: Simulated relative yield loss due to $O_3$ stress (solid lines) compared to the $O_3$ exposure relationships (dotted lines) from the literature data (symbols) for the (a) CERES-Maize, (b) CERES-Rice, (c) CROPGRO-Soybean, and (d) NWheat models. The $O_3$ exposure-yield response linear functions of the three $O_3$ sensitivities: tolerant (blue), intermediate (gold), and sensitive (magenta) are given in Figure S3. The cultivars were classified by grouping the cultivar**


**$O_3$ exposure-yield response (Fig. S2) into three evenly distributed quantiles: 66%-100%, 33%-66%, and 0%-33%, respectively. The $O_3$ sensitivities determined for each cultivar are listed in Table S10. The simulated results for the crop models use the $FOZ_1$ and $SFOZ_1$ values from Table 1. For each model, the same weather, soil, and dominant management conditions as in the normal rainfall and 350 ppm $CO_2$ treatment of the sensitivity analysis were used as reference (the $O_3$ response functions from the literature included $O_3$ field experiments conducted when the atmospheric $CO_2$ concentration**



**was ~350 ppm). The literature data consists of the relative yields (scaled to 25 ppb M7 O₃) of the cultivars examined in the Mills et al. (2018a) literature review combined with the maize and soybean cultivars used in this study for a total of 9 maize cultivars, 50 rice cultivars, 49 soybean cultivars, and 23 wheat cultivars (listed in Table S10). For the O₃ sensitivity of each crop, the root-mean-square error (RMSE) and coefficient of determination (r²) show the model performance compared to the linear fit of the O₃ exposure literature data (text color corresponds to O₃ sensitivity). The color shaded**
**area shows the standard error for the linear fit of the literature data for each of the cultivar O₃ sensitivities.**

## 4 Discussion

### 4.1 Simulating O₃ damage on crop yields

The measured yield losses for the maize FACE experiment were between 5% to 40% for the M7 O₃ concentrations when increasing from the ambient concentration (38 ppb) to the elevated O₃ treatment (77 ppb), a yield loss of 0.14%
to 1.01% per ppb M7 O₃ above the ambient concentration, depending on the O₃ cultivar sensitivity (Fig. 2b). Cv. NC338xHp301 and cv. Mo17xHp301 were classified as O₃ tolerant because of relatively small yield losses of 5% and 6%, respectively; cv. B73xMo17 was classified as O₃ intermediate with a yield loss of 11%; and cv. B73xHp301, cv. Mo17xNC338, and cv. B73xNC338 were sensitive to O₃ effects with yield losses of 22%, 30%, and 40%, respectively (Fig. S1a, Table S10). These cultivar O₃ sensitivities are based on a single experimental year so additional testing is
needed to further corroborate the classifications. Overall, the calibrated CERES-Maize model was able to reproduce these observed yield losses within 1%, i.e., simulated yield losses between 5% to 41%, or 0.12% to 1.05% per ppb O₃ increase above the ambient concentration. These yield losses were also calculated relative to 25 ppb (as described in section 2.5) for consistency with the literature, which resulted in simulated yield losses between 0.12% to 0.93% per ppb M7 O₃ increase above 25 ppb across the six cultivars.

When comparing the simulations to the maize O₃ exposure-yield relationships from the literature, the model simulated average yield losses of 0.16%, 0.36%, and 0.82% per ppb M7 O₃ increase above 25 ppb for the O₃ tolerant, intermediate, and sensitive cultivar O₃ sensitivities, respectively (Fig. 8 (a) solid lines). This agreed well with the literature yield losses of 0.24%, 0.33%, and 0.71% per ppb M7 O₃ increase above 25 ppb for the O₃ tolerant, intermediate, and sensitive cultivar sensitivities, respectively (Fig. S3 (a), Fig. 8 (a) dotted lines). The O₃ parameter
values used for the literature comparison were determined to provide the best fit across the literature experiments consisting of nine maize cultivars, but these O₃ parameter values could be calibrated for other scenarios and cases, i.e., higher or lower cultivar O₃ sensitivity.

The measured yield losses for the SoyFACE experiment were between 51% to 77% for the M7 O₃ concentrations when increasing from the ambient concentration (37 ppb) to the highest O₃ treatment (126 ppb) in 2009, a yield loss
of 0.57% to 0.86% per ppb M7 O₃ above the ambient concentration, depending on the cultivar O₃ sensitivity (Fig. 3 (c)). The calibrated CROPGRO-Soybean model reproduced observed yields losses within 10%, i.e., simulated yield losses between 59% to 80%, or 0.66% to 0.90% per ppb O₃ increase. Based on the calculated O₃ classifications from the literature and low yield divergence across the seven cultivars (Fig. S1 (b)), cv. Pioneer93B15, cv. Dwight, cv. IA-3010, and LN97-15076 were considered O₃ intermediate sensitivity, and cv. HS93-4118, cv. Loda, and cv. Pana were
considered O₃ sensitive (Table S10). In 2010, the observed soybean yield losses ranged between 31% to 76% when increasing from the ambient concentration (37 ppb) to the highest O₃ treatment (84 ppb), a yield loss of 0.65% to 1.60% per ppb M7 O₃ above the ambient concentration. The model underestimated yield losses in 2010, between 27%



to 44%, but because the experimental setup was the same for both years, an external factor may have affected yields that was not considered in the simulations (section 4.3). The 2010 yield losses were a similar magnitude to the 2009

yield losses, but the 2010 experiment had higher yield loss and variation per ppb $O_3$ increase with lower average M7 $O_3$ concentrations (Table 2, Fig. S4 (a)).

When comparing the simulations to the soybean $O_3$ exposure-yield relationships from the literature (Fig. 8 (c)), an average yield loss of 0.36%, 0.64%, and 0.96% per ppb M7 $O_3$ increase above 25 ppb was simulated for the $O_3$ tolerant, intermediate, and sensitive cultivar $O_3$ sensitivities, respectively. This was substantiated by the literature yield losses

of 0.45%, 0.63%, and 0.84% per ppb M7 $O_3$ increase above 25 ppb for the $O_3$ tolerant, intermediate, and sensitive cultivar $O_3$ sensitivities, respectively (Fig. S3 (c), Fig. 8 (c) dotted lines). The literature data consisted of 49 soybean cultivars, which had a smaller range of $O_3$ sensitivities compared to the other crops, although there were outliers where yield increased under higher $O_3$ concentrations (described in section 4.2).

The CERES-Rice model simulated an average yield loss of 0.05%, 0.23%, and 0.66% per ppb M7 $O_3$ increase above

25 ppb for the $O_3$ tolerant, intermediate, and sensitive cultivar $O_3$ sensitivities, respectively (Fig. 8 (b) solid lines). The rice literature had the most cultivars (50) of the four crops examined, and the simulated yield losses for the $O_3$ tolerant and intermediate cultivar $O_3$ sensitivities agreed well with the literature yield losses of 0.07% and 0.24% per ppb M7 $O_3$ increase above 25 ppb, respectively (Fig. 8 (b) dotted lines). A larger discrepancy between the simulated yield loss for the $O_3$ sensitive classification and the literature $O_3$ sensitive yield loss of 0.49% per ppb M7 $O_3$ increase above 25

ppb was due to the higher variability within the literature data (Fig. 8 (b) shaded area).

Using the calibrated NWheat model, the simulated yield losses were 0.26%, 0.66%, and 1.23% per ppb M7 $O_3$ increase above 25 ppb for the $O_3$ tolerant, intermediate, and sensitive cultivar $O_3$ sensitivities, respectively (Fig 8 (d)). These simulated yield losses were corroborated by the reported average yield losses of 0.33%, 0.61%, and 1.11% per ppb M7 $O_3$ increase above 25 ppb for the $O_3$ tolerant, intermediate, and sensitive cultivar $O_3$ sensitivities, respectively.

The literature expanded across different ranges of $O_3$ concentrations for all crops, and yield loss per ppb is not always constant over an expansive range of $O_3$ concentrations, so the model $O_3$ parameter values can be adjusted for higher or lower cultivar $O_3$ sensitivity.

### 4.2 Simulated relative yield loss with the combined effects of $O_3$, $CO_2$, and water deficit stress

The sensitivity analyses showed that the yield losses due to $O_3$ stress were higher under the normal rainfall and low

$CO_2$ treatment which agrees with previous literature that increased water availability increases $O_3$ impact due to increased stomatal uptake (Khan and Soja, 2003; Biswas et al., 2013). It was unexpected that the simulated $O_3$ photosynthetic response difference between the normal and reduced rainfall treatments for maize was less than 1% (Fig. 4 (c)). This was because the model simulated low water deficit stress under the 50% reduced rainfall treatment which obscured the $O_3$-water stress dynamics. Further reducing the rainfall to 40% of the normal amount increased

the simulated water deficit stress and produced the photosynthetic $O_3$-water dynamics consistent with the other models (Fig. S5). The elevated $CO_2$ concentration mitigated the detrimental effect of $O_3$ stress in the photosynthetic response for all models (Figs. 4 – 7 (c)), which agrees with recent global findings that elevated $CO_2$ concentrations can mitigate and even negate elevated $O_3$ impacts (Xia et al., 2021; Tai et al., 2021). Interestingly, the CROPGRO-Soybean model





simulated an inverse $O_3$-$CO_2$ effect on relative yield under the 50% rainfall condition when examining $SFOZ_1$ in detail
(Fig. 6 (d)). This inverse yield response was due to the low actual yield simulated under the 50% rainfall and low $CO_2$
treatment (< 2,000 kg ha$^{-1}$, Table S7) which resulted in smaller changes in yield compared to the 50% rainfall and
high $CO_2$ treatment, but the overall simulated aboveground biomass $O_3$-$CO_2$-water interaction was as expected (Fig.
S6).

For several of the observations from the actual soybean field experiment using cv. Pana, the yield increased under
higher $O_3$ concentrations (~2% to 18%, Fig. 3 (c) and Fig. S1 (b)). In some cases it is possible that elevated $O_3$
concentrations can benefit a crop via hormesis, a process where low levels of intermittent stress may benefit overall
crop growth through improved resiliency (Calabrese, 2014). It is also possible that if elevated $O_3$ concentrations reduce
biomass growth throughout the season, and therefore reduce nutrient resource demand throughout the season, small
yield increases can occur from a larger pool of resources available during the key reproductive/grain filling period
(Asseng and Van Herwaarden, 2003; Guarin et al., 2019). This increase in yield under higher $O_3$ concentrations was
also observed under several other soybean and rice cultivars from the literature (Fig. S2 (b) and (c)). However, a
soybean cultivar from the literature, cv. Cumberland, was reported to have a 34% increase under elevated $O_3$ (67 ppb)
compared to the control treatment (25 ppb), but such a large increase may indicate that another outside factor affected
the yields.

### 4.3 Uncertainty in model simulations and $O_3$ exposure field experiments

Crop models contain uncertainties due to simplification of complex biological processes, but field experiments may
also contribute uncertainty via measurement. The soybean simulations overestimated both biomass and yield across
all cultivars and treatments for the 2010 SoyFACE experiment. Since both the ambient and elevated $O_3$ treatments
were overestimated, it is unlikely that the simulated $O_3$ interactions caused the discrepancy. Examining the weather
input showed a 14% increase in cumulative incoming solar radiation for the 2010 growing season compared to the
2009 growing season (Fig. S4 (b)). The 2010 season was warmer than the 2009 season, average seasonal temperature
of 23.4 °C compared to 19.1 °C, but no heat stress was reported and the difference in rainfall was negligible, 445 mm
compared to 454 mm. Since management was the same for both years and no water or N stresses were reported, it was
expected that the 2010 yields would be higher than the 2009 yields due to the increased solar radiation, but the average
2010 yield across all cultivars for the ambient treatment decreased, 3300 kg ha$^{-1}$ in 2010 compared to 3700 kg ha$^{-1}$ in
2009. Therefore, it is possible that an outside stress factor not considered within the model limited soybean growth in
the field in 2010 which led to the model overestimating biomass and yield.

The sensitivity analyses showed that the $CO_2$ effect was more pronounced in the model photosynthesis response than
in the leaf senescence response (compare solid and dashed lines in Figs. 4 – 7 (c) and (d)). This is because the models
do not have a $CO_2$ effect directly applied to the daily leaf senescence calculation, whereas $CO_2$ directly affects the
daily photosynthesis calculation (PCARB in Eq. (3) and (4), and PRATIO in Eq. (9)). Improved $CO_2$ representation
within the crop models is being explored through the Agricultural Model Intercomparison and Improvement Project
(AgMIP) studies (Ahmed et al., 2017; Ahmed et al., 2019; Toreti et al., 2020), but additional high-quality data is
needed for model testing.



## 5 Conclusion


Crop responses to elevated $O_3$ concentrations were incorporated into the DSSAT CERES-Maize, CERES-Rice, CROPGRO-Soybean, and NWheat crop models via functions reducing photosynthetic activity and accelerating leaf senescence. Model testing showed that each of the four models reproduced the observed $O_3$ response from field experiments and previous literature, as well as the expected interactions between $O_3$, $CO_2$, and water deficit stress.


This incorporation allows for improved simulation of the heterogeneity of $O_3$ impacts across geographical regions and systems, as well as across years within seasons, which is more representative of real-world interactions than using a generic damage coefficient. Overall, increasing M7 $O_3$ concentrations had a negative effect on growth and yield across all four crops, and this negative effect was exacerbated by increased water availability and ameliorated by elevated $CO_2$ concentrations. The $O_3$ impact and stress response of the crop depends on the stress severity, duration, frequency,


cultivar sensitivity, and seasonal timing (i.e., developmental stage) which can be accounted for by using the updated crop models.

The addition of $O_3$ stress functionality into crop models will improve simulations of global environmental interactions using a key factor that is often not included in agricultural and climate change assessments. The DSSAT models in this study can be used to simulate the $O_3$ impacts on crops in combination with climate change. To further improve


model performance, the models should continue to be tested with additional experimental data and compared with other $O_3$-modified crop models as part of multi-model ensemble assessments conducted by the AgMIP (https://agmip.org/). The framework described here can be used by other process-based crop models, local or gridded, to incorporate $O_3$ stress interactions into the model. This model improvement also suggests potential future collaboration between crop modelers and remote sensing experts using weather and climate models with dynamic


chemistry components, such as the NASA Atmosphere Observing System (https://aos.gsfc.nasa.gov/).

### Code availability

The current version of the DSSAT crop modeling platform is available to download from the DSSAT Foundation website (https://dssat.net/). The current version of the pSIMS framework is available to download from the RDCEP website (http://www.rdcep.org/research-projects/psims). The $O_3$-modified version of the DSSAT crop models will be


available with the next DSSAT version release, and the $O_3$-modified version of the pDSSAT crop models is available from the GitHub repository at https://github.com/jguarin4/dssat-csm-os/tree/develop_v4.8_pdssat. An archived version of the code is also available on Zenodo at https://zenodo.org/badge/latestdoi/232137043.The R code used to classify the cultivar $O_3$ sensitivities is available on the Harvard Dataverse at https://doi.org/10.7910/DVN/0NN9MH.

### Data availability


All field experimental and literature data used in this study are available from the sources referenced. The crop model simulated output data is available on the Harvard Dataverse at https://doi.org/10.7910/DVN/0NN9MH.



**Author Contribution**

J.R.G. and J.J. designed and conducted the study. E.A.A. provided the $O_3$ exposure field data. K.S. collated the $O_3$ exposure literature data. J.R.G. and F.O. incorporated the $O_3$ modifications into the DSSAT/pDSSAT model code. 595 S.A., K.B., L.E., G.H., and A.C.R. provided insight on $O_3$-crop interactions within the crop models. J.E., I.F., and D.K. provided technical support and guidance for the pSIMS/pDSSAT framework. J.R.G. and J.J. co-wrote the manuscript. All authors contributed to editing the manuscript.

**Competing interests**

The authors declare that they have no conflict of interest.

**Acknowledgements**

The authors would like to thank Amy Betzelberger and Nicole Choquette for sharing the $O_3$ field experiment data. J.R.G. and K.S. would like to thank Stephanie Osborne for help with collecting the $O_3$ exposure literature data. J.R.G. and J.J. were supported by the Open Philanthropy Project. J.R.G., J.J. and A.C.R. contributions were also enabled by NASA Earth Science Division support of the NASA GISS Climate Impacts Group.

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
