# Peer review of "Modeling the effects of tropospheric ozone on the growth and yield of global staple crops with DSSAT v4.8.0"

_EGUsphere, 2023_

## Author Comment (AC1)

**Letter of Response (responses below each comment, track changes manuscript attached)**

**Reviewer #1 (R1):**

**R1:** The authors tended to enable DSSAT is able to simulate 1) how O3 affects photosynthesis and leaf senescence, and 2) how O3 effect interacts with CO2 and water stress by associated changes in stomatal conductivity for crops. My comments are as follows:

(1) All stress factors the authors newly incorporated into model are all for the effects of photosynthesis (like FO3) and lead senescence (like SLFO3). But I noticed that the model parameterizations are based on only relative yields (Fig. S1-S3). It is important to have photosynthesis and leaf observations as you tend to simulate their changes to validate model, rather than fitting yields by giving different combinations of parameters. Otherwise, it is very possible you have a "right" yield simulation but wrong parameters, which will give users a lot of trouble when they tend to project model to some unknown conditions.

**Response:** We agree on the importance of simulating the crop response correctly to avoid getting the "right" yield for the wrong reasons. The O3 stress routines were incorporated into the DSSAT models following the existing framework of abiotic stress routines on the photosynthetic and leaf senescence crop growth processes, which has been tested and applied for other stresses (Asseng et al., 2004). The DSSAT crop phenology and growth parameters are based on not only the relative yield loss due to O3, but also on the observed phenology, biomass, and yield data. We first calibrated these parameters for the observed control treatment (negligible/minimal O3 effect) using all experimental observations applicable to the model output to ensure that the models were functioning properly regardless of O3 impact, i.e., to ensure that we were getting the "right" yield for the right reasons. We then calibrated the FOZ1 and SFOZ1 response parameters to simulate the relative yield loss due to O3 stress while keeping the control crop parameters so that they correspond to the observed O3 effect.

We now clarify this in lines 105-108, "*The incorporation of $O_3$ effects into the DSSAT crop models followed the same methodology as the $O_3$ incorporation into the DSSAT-NWheat crop model (Guarin et al., 2019), which was based on the incorporation of previous abiotic stress routines (Asseng et al., 2004).*"

And we now added in lines 222-225, "*For each crop, the DSSAT phenological and growth parameters were calibrated based on the observed control treatment with minimal $O_3$ stress to ensure that the models were functioning properly regardless of $O_3$ impact. Then, the $O_3$ response parameters, $FOZ_1$ and $SFOZ_1$, were calibrated based on the observed $O_3$ exposure-yield response between the elevated $O_3$ treatments and the control to simulate the $O_3$ effect.*"

Since photosynthesis in the models is driven by radiation use efficiency (RUE), it would be helpful to gather observations of RUE under different O3 treatments for further comparison, but detailed O3 experiment data is limited and that was outside of the scope of the study. With improved observational data, it would be an important follow-up study to strengthen the robustness of the O3 stress routines. We now mention the benefit of additional model testing in the concluding remarks in lines 602-606, "*The $O_3$ parameter values in this study can be used as preliminary approximations, but to further improve model performance and robustness of the $O_3$ stress routines, the models and parameters should continue to be tested and calibrated with additional $O_3$ exposure experimental data when available. In addition, the models should be*

*compared with other O₃-modified crop models as part of multi-model ensemble intercomparison and improvement assessments conducted by the AgMIP ([https://agmip.org/](https://agmip.org/)).*"

*Asseng, S., Jamieson, P. D., Kimball, B., Pinter, P., Sayre, K., Bowden, J. W., and Howden, S. M.: Simulated wheat growth affected by rising temperature, increased water deficit and elevated atmospheric CO2, Field Crops Research, 85, 85-102, 10.1016/s0378-4290(03)00154-0, 2004.*

**R1:** (2) In line 260, "until the best fit was found for the phenology, growth, and relative yield loss for each cultivar across all O3 treatments." Where are the "growth" observations? I can only see relative yield data.

**Response:** The growth observations referred to the maize yield observations in Fig. 2a and the soybean biomass and yield observations in Fig. 3a/b/d/e. We now clarify the sentence as, "*...until the best fit was found for the phenology, aboveground biomass and yield, and relative yield loss for each cultivar across all O3 treatments.*"

**R1:** (3) For their experiment, I noticed some odd observations in Fig. S2. In rice and soybean, I found yields of some cultivars increase with higher O3, which seems impossible for the new models except unrealistic parameters were fed into (like change some parameters from negative to positive to make it increase rather than decrease. But this is inconsistent with the theory the models were built). Please explain the odd observations in the main text.

**Response:** We explained the potential reasons for the 'odd' observations in the discussion in lines 550-560. In rare cases it is possible for crop growth to increase under elevated O3, e.g., due to hormesis or changes in seasonal nutrient dynamics (i.e., reduced seasonal biomass resulting in higher nutrient pools during grain filling). However, the large increase seen in the soybean cv. Cumberland may be anomalous due to outside/unobserved factors. The experimentalists from that soybean study speculated that the difference may have been due to changes in the water relations affecting the amount of drought stress, but it was not measured.

We now clarify this in the text in lines 560-562, "*The experimentalists speculated that the large yield difference was due to changes in the seasonal water dynamics thereby causing increased drought stress under the control treatment compared to the elevated O₃ treatment (Mulchi et al., 1988).*"

*Mulchi, C. L., Lee, E., Tuthill, K., and Olinick, E. V.: Influence of ozone stress on growth-processes, yields and grain quality characteristics among soybean cultivars, Environmental Pollution, 53, 151-169, 10.1016/0269-7491(88)90031-0, 1988.*

**R1:** (4) I appreciate for the modelers consider the interactions between O3, CO2 and water stress by stomatal conductivity, which I am really interested into. I wish the authors could add more observation points in Fig 4 and 5 to ensure the model can simulate the key interactions quantitively.

**Response:** Thank you for the positive comments. We would also prefer to include observations in Fig. 4 and 5, but detailed O3 experimental data is limited and we do not have observational data for the simulated sensitivity analysis scenarios. However, to further support that the models can simulate key interactions, we now compare the model performance to the performance of the well-known Weibull O3 response functions (added in Table S11) in Fig. 8. Overall, the crop models performed better than the Weibull damage response functions for every crop and O3 classification, except the O3 intermediate classification for soybean (although the O3 intermediate soybean RMSE difference was < 1%, RMSEsim = 2.4% vs RMSEweibull = 1.9%).

We now added some discussion on the crop model and Weibull function performance in lines 527-534, "*As an additional check of model performance, the calculated relative yield from the well-known Weibull O₃ response functions (Table S11) were compared to the literature O₃ exposure linear yield responses for each crop and O₃ classification (Fig. 8). The Weibull function performance was then compared to the simulated crop model results. Overall, the crop model simulations performed better (lower RMSE and higher $r^2$) than the Weibull response functions across all crops for all three O₃ classifications, except the O₃ intermediate classification for soybean which had < 1% difference between the RMSE (compare RMSE and $r^2$ in Fig. 8). The performance results suggest that it is best to use calibrated crop models when available, and that the Weibull response functions are mainly representative of O₃ intermediate classifications for maize, rice, and soybean, and O₃ tolerant classifications for wheat.*"

**Reviewer #2 (R2):**

**R2:** This is an impressive compilation of data for many crops from multiple sites to calibrate the data intensive crop models. This was an enormous amount of work. Overall, this is a good paper that highlights the impacts of ozone exposure on crop yields along with other stressors and should be of interest to a wide audience.

**Response:** Thank you for the positive comments.

**R2:** The work focuses on the M7 (7-hour daily mean) ozone metric to alter daily photosynthesis and accelerate leaf senescence. The authors claim that their approach is more representative than a generic annual damage function. It would have been nice to see a comparison of a simpler damage function approach with the simulation models presented in this paper to compare the results of the two approaches. The use of the models outlined in this paper may be difficult to apply because of the need for large amounts of detailed data for each site. Using weighted seasonal metrics like AOT40, W126 or SUM06 to modify yield might produce robust results as well. However, I realize the M7 metric for yield loss is the most readily available in the literature.

**Response:** We agree that comparing the model results to a separate damage function would be informative. As proposed, we updated Figure 8 to compare the literature data O3 classifications to the well-known Weibull O3 damage response functions for each crop (Weibull equations added in supplementary Table S11). We then compared the RMSE and r2 between the model simulations and the Weibull functions, which highlights that the process-based crop models consistently performed better than the empirical damage functions for every crop O3 classification, except the O3 intermediate classification for soybean (although the O3 intermediate soybean RMSE difference was < 1%, RMSEsim = 2.4% vs RMSEweibull = 1.9%).

We now added some discussion on the crop model and Weibull function performance in lines 527-534, "*As an additional check of model performance, the calculated relative yield from the well-known Weibull O₃ response functions (Table S11) were compared to the literature O₃ exposure linear yield responses for each crop and O₃ classification (Fig. 8). The Weibull function performance was then compared to the simulated crop model results. Overall, the crop model simulations performed better (lower RMSE and higher $r^2$) than the Weibull response functions across all crops for all three O₃ classifications, except the O₃ intermediate classification for soybean which had < 1% difference between the RMSE (compare RMSE and $r^2$*

*in Fig. 8). The performance results suggest that it is best to use calibrated crop models when available, and that the Weibull response functions are mainly representative of O₃ intermediate classifications for maize, rice, and soybean, and O₃ tolerant classifications for wheat.*"

And in the conclusion, we added lines 591-592, "*The simulated yield responses were also more representative of the O₃ exposure literature data than the well-known Weibull O₃ response functions for all crops.*"

Considering other O3 seasonal metrics like AOT40, W126, or SUM06 would be interesting, but as mentioned, we used the M7 metric because it is the most readily available in the literature. There are also conversion functions available to convert between M7, AOT40, M12, and M24 (Osborne et al., 2016). The incorporation of M7 into the DSSAT models allows for comparison to other crop models that use different ozone damage metrics, e.g., DO3SE model, as part of multi-model intercomparison studies. This can be an interesting area for future research. We now mention the O3 metrics in lines 125-128, "*The M7 O₃ metric was chosen as the model input because it is the most readily available metric in the literature, and conversion functions exist to convert between M7 and AOT40, daily mean 12-hour (M12), or daily mean 24-hour (M24) O₃ metrics (Osborne et al., 2016).*"

*Osborne, S. A., Mills, G., Hayes, F., Ainsworth, E. A., Buker, P., and Emberson, L.: Has the sensitivity of soybean cultivars to ozone pollution increased with time? An analysis of published dose-response data, Global Change Biology, 22, 3097-3111, 10.1111/gcb.13318, 2016.*

**R2:** The models are well thought-out and carefully constructed. However, it is hard to know how the models will work in uncalibrated situations. The only predictive modeling seems to have been done on the evaluation year, 2010, at the SoyFACE study in Illinois calibrated with the 2009 SoyFACE study. More attention to investigating why the simulations overestimated biomass and yield in 2010 could have been presented. For example, even though the rainfall at that site was similar between years, 2010 appears to have more of the rainfall at the beginning of the season compared to rainfall patterns in 2009 (Betzelberger et al., 2012). And the mass of individual seeds were smaller at lower ozone exposures in 2010.

**Response:** Thank you for pointing out the differences in the early season rainfall. The effects of excessive moisture, such as flooding or water logging, are underrepresented in the current generation of DSSAT crop models, it is possible that this may have affected the seed germination or emergence of the crop which may have led to the model overestimation in 2010. We examined this in more detail to find that there was 221 mm of rainfall in the first 30 days of the 2010 season, higher than the 153 mm in the 2009 season, and this may have affected crop growth. To clarify this, we added the cumulative rainfall plot in Figure S4 (c) in the supplementary and updated lines 575-578, "*One possibility is that increased rainfall during the beginning of the 2010 season (221 mm in first 30 days compared to 153 mm in first 30 days of 2009 season, Fig. S4 (c)) may have resulted in germination or emergence stress due to excessive water such as flooding or lodging, which are factors not yet considered in the crop models.*"

We also now mention the benefit of additional model testing and calibration in lines 602-606, "*The O₃ parameter values in this study can be used as preliminary approximations, but to further improve model performance and robustness of the O₃ stress routines, the models and parameters should continue to be tested and calibrated with additional O₃ exposure experimental data when available. In addition, the models should be compared with other O₃-modified crop models as*

*part of multi-model ensemble intercomparison and improvement assessments conducted by the AgMIP ([https://agmip.org/](https://agmip.org/)).*"

**R2:** I am not an expert in the modeling field or the uses of this model. I would have liked to have seen more discussion the implications of this model. How can it be used in the near-term? Do users of the model have to have access to detailed site data or can modeled parameters be used to drive the model?

**Response:** As suggested, we have expanded the concluding remarks on near-term applications for the model and model use in lines 599-600, "*The addition of $O_3$ stress functionality into crop models will improve both near- and long-term simulations of global environmental interactions using a key factor that is often not included in agricultural and climate change assessments.*

And we now added lines 607-609, "*As a next step, the AgMIP Ozone team is currently conducting a multi-model ensemble study with crop models that have the capacity to evaluate the responses of future crop yields to different ozone concentrations. This effort will help produce more robust estimates of climate change impacts in global agriculture.*"

We also clarified now the application of the model parameters and the benefit of detailed site data in lines 602-604, "*
[revised manuscript text omitted]

---

## Author Response (AR2)

**Letter of Response (responses below each comment)**

**Reviewer #1 (R1):** Thanks for the revision of authors. After reading, I still have one concerns:

For the odd observations, the authors claimed an increased drought effects in control treatment. It seems that the new model has reproduced this confounding effect, as the model has simulated a higher yield for elevated O3 in these experiments. I think this is a great example to show the advantage of DSSAT model. Please show the simulation results in the manuscript (like the simulated soil moisture between the two treatments).

**Response:** There may be a misunderstanding with the description of the odd observations. The sentence detailing the increased drought effects on the control treatment in lines 558-560 was in reference to the soybean anomalous observation (34% increase in cv. Cumberland) from Mulchi et al., 1988 mentioned in lines 554-558. To clarify this, we modified the sentence in lines 558-560 to, "*Mulchi et al. (1988) speculated that the large yield difference reported was due to changes in the seasonal water dynamics thereby causing increased drought stress under the control treatment compared to the elevated O3 treatment.*"

For the soybean field experiment that was used for model calibration, only one of the seven cultivars, cv. Pana, showed an odd increase in yield as O3 increased (mentioned in lines 548-549). Although the exact cause of this is unknown, we describe the potential causes in lines 549-555, i.e., hormesis or increases in available end-of-season resources due to seasonal nutrient dynamics. Regarding model performance for this cultivar, the model did not reproduce the yield increase under higher O3 reported in four of the nine treatments (see Fig. S7 below), however, the simulated results were still acceptable because the model performed well in the other five treatments where elevated O3 decreased yield (RMSE = 0.14, r2 = 0.887). It may be possible for the models to simulate yield increases under elevated O3 depending on the interactions of other stresses and available resources throughout the season which was seen in Guarin et al. (2019) Fig. 5a.

[Figure]

**Figure S7: Observed relative yield under elevated M7 O₃ concentrations (left) and simulated model performance of the relative yield (right) of the soybean cultivar, cv. Pana (PA). The root-mean-square error (RMSE) and coefficient of determination (r²) show the model performance. Solid black line shows 1:1 comparison and dotted black line shows linear fit.**

We now added Figure S7 to the supplementary and reference it in lines 548-549, "*For several of the observations from the actual soybean field experiment using cv. Pana, the yield increased under higher O₃ concentrations (~2% to 18%, Fig. 3 (c), Fig. S1 (b), and Fig. S7)*".

To clarify the model performance, we added the sentences in lines 560-563, "*Reproducing rare occurrences where elevated $O_3$ may result in yield increases can be a challenge for the models because of the linear response of the stress equations (Fig. S7). However, it may be possible depending on the simulated interactions between seasonal dynamics of resources as shown with the sensitivity analysis of wheat yields in Guarin et al. (2019).*"